# Machine Learning Analysis of the Impact of Silver Nitrate and Silver Nanoparticles on Wheat (*Triticum aestivum* L.): Callus Induction, Plant Regeneration, and DNA Methylation

**DOI:** 10.3390/plants12244151

**Published:** 2023-12-13

**Authors:** Aras Türkoğlu, Kamil Haliloğlu, Fatih Demirel, Murat Aydin, Semra Çiçek, Esma Yiğider, Serap Demirel, Magdalena Piekutowska, Piotr Szulc, Gniewko Niedbała

**Affiliations:** 1Department of Field Crops, Faculty of Agriculture, Necmettin Erbakan University, Konya 42310, Türkiye; 2Department of Field Crops, Faculty of Agriculture, Ataturk University, Erzurum 25240, Türkiye; kamilh@atauni.edu.tr; 3Department of Agricultural Biotechnology, Faculty of Agriculture, Igdır University, Igdir 76000, Türkiye; fatih.demirel@igdir.edu.tr; 4Department of Agricultural Biotechnology, Faculty of Agriculture, Ataturk University, Erzurum 25240, Türkiye; maydin@atauni.edu.tr (M.A.); semra.cicek@atauni.edu.tr (S.Ç.); esma.yigider@atauni.edu.tr (E.Y.); 5Department of Molecular Biology and Genetics, Faculty of Science, Van Yüzüncü Yıl University, Van 65080, Türkiye; serapdemirel@yyu.edu.tr; 6Department of Geoecology and Geoinformation, Institute of Biology and Earth Sciences, Pomeranian University in Słupsk, 27 Partyzantów St., 76-200 Słupsk, Poland; magdalena.piekutowska@upsl.edu.pl; 7Department of Agronomy, Poznań University of Life Sciences, Dojazd 11, 60-632 Poznań, Poland; piotr.szulc@up.poznan.pl; 8Department of Biosystems Engineering, Faculty of Environmental and Mechanical Engineering, Poznań University of Life Sciences, Wojska Polskiego 50, 60-627 Poznań, Poland

**Keywords:** artificial intelligence, genetic algorithm, in vitro culture, modeling, prediction, wheat

## Abstract

The objective of this study was to comprehend the efficiency of wheat regeneration, callus induction, and DNA methylation through the application of mathematical frameworks and artificial intelligence (AI)-based models. This research aimed to explore the impact of treatments with AgNO_3_ and Ag-NPs on various parameters. The study specifically concentrated on analyzing RAPD profiles and modeling regeneration parameters. The treatments and molecular findings served as input variables in the modeling process. It included the use of AgNO_3_ and Ag-NPs at different concentrations (0, 2, 4, 6, and 8 mg L^−1^). The in vitro and epigenetic characteristics were analyzed using several machine learning (ML) methods, including support vector machine (SVM), random forest (RF), extreme gradient boosting (XGBoost), k-nearest neighbor classifier (KNN), and Gaussian processes classifier (GP) methods. This study’s results revealed that the highest values for callus induction (CI%) and embryogenic callus induction (EC%) occurred at a concentration of 2 mg L^−1^ of Ag-NPs. Additionally, the regeneration efficiency (RE) parameter reached its peak at a concentration of 8 mg L^−1^ of AgNO_3_. Taking an epigenetic approach, AgNO_3_ at a concentration of 2 mg L^−1^ demonstrated the highest levels of genomic template stability (GTS), at 79.3%. There was a positive correlation seen between increased levels of AgNO_3_ and DNA hypermethylation. Conversely, elevated levels of Ag-NPs were associated with DNA hypomethylation. The models were used to estimate the relationships between the input elements, including treatments, concentration, GTS rates, and *Msp I* and *Hpa II* polymorphism, and the in vitro output parameters. The findings suggested that the XGBoost model exhibited superior performance scores for callus induction (CI), as evidenced by an R^2^ score of 51.5%, which explained the variances. Additionally, the RF model explained 71.9% of the total variance and showed superior efficacy in terms of EC%. Furthermore, the GP model, which provided the most robust statistics for RE, yielded an R^2^ value of 52.5%, signifying its ability to account for a substantial portion of the total variance present in the data. This study exemplifies the application of various machine learning models in the cultivation of mature wheat embryos under the influence of treatments and concentrations involving AgNO_3_ and Ag-NPs.

## 1. Introduction

Plant tissue culture, also referred to as in vitro culture, is a method of cultivating plant cells or plant parts in a controlled, sterile, artificial environment on a nutrient medium. It holds significant importance in various realms of plant biology, encompassing embryogenesis, morphogenesis, cytology, nutrition, germplasm conservation, extensive clonal propagation, genetic engineering, pathology, and the generation of pathogen-free plants and valuable metabolites [1]. In vitro plant culture is influenced by factors such as genotype, the composition of the culture medium, and the presence of plant growth regulators.

Plant growth regulators (PGRs) play a crucial role in governing growth, differentiation, shoot initiation, proliferation, callus induction, embryogenesis, and root development in in vitro studies. The five classes of PGRs that regulate plant growth are auxins, gibberellins, cytokinins, ethylene, and abscisic acid. Each category primarily includes both synthetic and naturally occurring compounds. Calluses and plantlets, generated through plant tissue and cell culture, contribute to the release of ethylene. Ethylene, a plant growth regulator, is known to influence morphogenesis in plant culture [2,3,4]. Prior research has shown that ethylene has the potential to hinder shoot rejuvenation, callus growth, and somatic embryogenesis [5,6]. Closed vessels are utilized in tissue culture to prevent contamination. However, in certain instances, this practice can lead to atypical plant development due to the accumulation of gases, such as ethylene, within the vessels used for tissue culture. Researchers have sought to mitigate the negative impact of ethylene on the regeneration process in two ways: by removing the gas from the air through mechanical ventilation, and by employing chemical substances to either limit its production or impede the hormone’s function [7,8]. The incorporation of ethylene antagonists in the culture media distinctly influences the concentration of ACC (1-amino cyclopropane-1-carboxylic acid), leading to consequential alterations in ethylene levels [9]. Furthermore, the introduction of silver ions into the culture medium, specifically in the form of silver nitrate (AgNO_3_), which acts as an inhibitor of ethylene, has been demonstrated as a successful method for enhancing the regeneration process and subsequently increasing the likelihood of transformation. This approach has been observed to have a significant impact on somatic embryogenesis and the development of shoots in various plant tissues under in vitro conditions [2,10]. Various perspectives and examples of experimental evidence have been presented to explain how silver ions can obstruct ethylene receptors, rendering plants resistant to ethylene [2,5,11].

Nanotechnology, a branch of science, involves the manipulation of materials at the atomic level, typically at a scale smaller than 100 nanometers, to enhance their functionality [12]. Its applications span various fields, including the degradation and distribution of pesticides, the development of nanosensors, the utilization of micronutrients in agriculture, and plant protection and nutrition [13]. Nanotechnology offers viable approaches for safeguarding soil health and conditions by reducing agricultural waste and mitigating environmental contamination [14,15]. Nanoparticles (NPs) have the potential to significantly enhance the functioning of agriculture [14]. Due to their high surface area-to-volume ratio, nanoparticles are highly biologically active [16]. 

These characteristics endow nanotechnology with immense commercial potential, but they also underscore various health and environmental concerns [17]. Nanoparticles (NPs) have demonstrated effectiveness in enhancing the regeneration, morphological development, morpho-physiology, and biochemical parameters of in vitro-produced plantlets [18]. Silver nanoparticles (Ag-NPs) have garnered significant attention, owing to their well-established antimicrobial properties. These properties have been effectively utilized in various medical applications, devices, textiles, clothing, and food packaging, as well as healthcare and household products [19]. In the pursuit of enhancing the quality of tissue-cultured plantlets through various means, silver nanoparticles (Ag-NPs) have been a subject of intense research. Recent studies show that Ag-NPs have been employed for various applications, including reducing microbial contamination [20], inducing somaclonal variation [21], enhancing proliferation rates [22], and in vitro bioactive chemical generation [23]. Therefore, this study represents an initial endeavor to examine the influence of AgNO_3_ and Ag-NP treatments on in vitro wheat culture, with a specific emphasis on callus induction and regeneration efficiency. Wheat, a paramount crop in temperate regions, holds the distinction of occupying the largest cultivable area globally [24]. Hence, studies on wheat continue to receive significant attention due to its paramount importance in human nutrition, its adaptability to a wide range of climates, and its abundance of essential nutrients [25].

Several studies have been conducted to investigate the effects of nanoparticles (NPs) on cytosine methylation in human cell DNA [26,27,28,29]. However, limited research exists on the epigenetic modifications induced in plant DNA by nanomaterials. In recent studies, it has been proposed that nanoparticles have the potential to induce epigenetic alterations in plants, such as cytosine DNA methylation and histone modification [30,31,32,33,34]. DNA meth-ylation occurs when a methyl group from S-adenosyl-L-methionine is transferred to the carbon five position of cytosine, resulting in 5-methylcytosine (5-meC) [35]. Gene expres-sion patterns can be influenced by DNA methylation, miRNA (microRNA), and retrotransposon activities, all of which have the potential to contribute to genomic insta-bility [36]. Recently, alterations in DNA methylation have been detected using various methods, including coupled restriction enzyme digestion–random amplification (CRED-RA) [37] and coupled restriction enzyme digestion–inter-primer binding site (CRED-iPBS) [38] methylation-sensitive amplified fragment length polymorphism (met-AFLP) [39], DArTseqMet [40], methylRAD [41], methyl-seq [42], and semiquantitative MSAP [43]. The objective of our study was to investigate alterations in cytosine methyla-tion in wheat when exposed to silver nitrate and Ag-NPs in an in vitro condition using the CRED-RA technique. 

Research on plant tissue culture explores the impact of various input parameters, whether singular or numerous, on the regenerative capacity of targeted plant species [44]. In most cases, traditional statistical methods have been employed to analyze and interpret the output variables. These methodologies often utilize variance analysis and linear regression models to determine the degree of correlation between independent input factors and dependent output variables [45,46]. Complex factors and non-linear features present some of the most significant challenges for researchers working in the field of plant tissue culture [47,48]. Addressing these challenges may be achieved through the utilization of effective high-throughput technologies, such as machine learning (ML) models. These models enable the examination and enhancement of output variables in relation to input factors [49,50]. The utilization of various machine learning algorithm models in the field of plant biotechnology is a growing and emerging area of research. This area focuses on the prediction and optimization of variables in complex biological systems [51,52]. Ma-chine learning (ML) is a well-recognized framework within the field of data science that addresses complex difficulties across various scientific fields. This methodology surpass-es traditional one-way analyses, enabling a more nuanced understanding and accurate interpretation of findings [53,54]. The application of machine learning has demonstrated favorable results in various areas of plant science, including in vitro germination [49,51], regeneration studies [49,55], and mutagenesis [56]. 

The application of mathematical frameworks and artificial intelligence (AI)-based models under in vitro conditions is still notably limited to wheat. The goal is to understand the complex dynamics of callus induction, regeneration efficiency, and the DNA methylation process. Within the framework of this specific scenario, the goals of this research were threefold: (1) examining the effects of AgNO_3_ and Ag-NP treatments on wheat in vitro regeneration; (2) identifying RAPD profiles and DNA methylation changes among experimental groups; and (3) modeling the observed in vitro regeneration parameters of wheat using all complex inputs and formulating the predicted results.

## 2. Results

### 2.1. Effects of Silver Nitrate and Silver Nanoparticles on In Vitro Parameters

The analysis of variance demonstrated that the types of treatment (AgNO_3_ and Ag-NPs) were not significant for the whole set of investigated parameters (CI, EC, and RE). Among the experimental groups, it was determined that the concentrations of AgNO_3_ and Ag-NPs did not provide statistically significant results for CI (Table 1). However, these concentrations did exhibit statistical significance for EC (*p* ≤ 0.01) and RE (*p* ≤ 0.05). Based on the obtained variance result pertaining to the interaction between the treatment and concentration, it was concluded that a statistically significant difference was observed across all attributes (*p* ≤ 0.001) (Table 1).

The MS medium with 2 mg L^−1^ Ag-NPs had the maximum EC rate, which was 95.00%, while the MS medium with 4 mg L^−1^ AgNO_3_ had the lowest value, which was 60%. The EC ratios obtained from MS medium with 2 mg L^−1^ Ag-NPs and 4 mg L^−1^ AgNO_3_ concentrations ranged from 85.0% to 37.5%, representing the greatest and lowest values, respectively. The MS medium containing 8 mg L^−1^ AgNO_3_ exhibited the highest recorded RE value of 0.83, whilst the MS medium containing 4 mg L^−1^ AgNO_3_ had the lowest recorded value of 0.13 (Table 1, Figure 1 and Figure 2).

Analysis revealed that the EC parameter exhibited the maximum value of 77.50% at an average concentration of 2 mg L^−1^, while the RE parameter showed the highest value of 0.69 at an average concentration of 6 mg L^−1^. The results of this study demonstrate that varying concentrations of Ag-NPs have a notable influence on the development of plants, with certain concentrations of Ag-NPs exhibiting the potential to improve plant growth.

### 2.2. RAPD Analysis

A RAPD analysis was conducted to assess the polymorphic effect of simultaneous treatment with varied concentrations of AgNO_3_ and Ag-NPs on wheat genomic DNA (Figure 3). An evaluation of the number of polymorphic bands generated by the applied primers was conducted by comparing control plants to the plants treated with AgNO_3_ and Ag-NPs (Table 2). The control exhibited a total of 29 bands, with the OPW-6 marker generating the highest number of bands (seven bands), and the OPH-17 marker generating the lowest number of bands (one band). The polymorphic bands ranged from 207 base pairs (bp) to 812 bp (for OPH-17 and OPW-4, respectively). Furthermore, the experimental group exposed to 8 mg L^−1^ AgNO_3_ displayed the smallest molecular size (207 bp), while the experimental groups exposed to 4 mg L^−1^ Ag-NPs and 6 mg L^−1^ Ag-NPs showed the largest molecular size (812 bp). Following the administration of varying doses, the experiments treated with AgNO_3_ and Ag-NPs exhibited significant alterations in their RAPD profiles. In comparison to the control group, the experimental groups showed a significant net increase of 51 newly created bands and a notable net loss of 19 pre-existing bands (Figure 4A). The observed polymorphism rates varied between 20.7% (at a concentration of 2 mg L^−1^ AgNO_3_) and 44.8% (at a concentration of 8 mg L^−1^ Ag-NPs) (Figure 4B). A measurement of alterations in the RAPD profiles was conducted using the genomic template stability (GTS) percentage. The treatment with a concentration of 2 mg L^−1^ AgNO_3_ exhibited the greatest GTS value (79.3%), while the treatment with a concentration of 8 mg L^−1^ Ag-NPs showed the lowest value (55.2%). The GTS value was at its lowest during the treatment involving AgNO_3_ at a concentration of 6 mg L^−1^, as well as during the treatment using Ag-NPs at a concentration of 8 mg L^−1^ (Figure 4C).

### 2.3. CRED-RA Analysis

The outcomes of the CRED-RA test are presented as the proportion of polymorphism observed in the CRED-RA assays that underwent digestion via *Msp I* and *Hpa II* enzymes (Figure 5 and Table 3). These findings indicate that the DNA methylation status, whether hypermethylated or hypomethylated, depended on the type and quantity of treatments (AgNO_3_ and Ag-NPs). This determination was made by comparing the PCR products obtained from the control DNA to those obtained from the experimental DNA. The data obtained reveal that the *Msp I*- and *Hpa II*-digested controls exhibited 77 and 73 bands, respectively. The total count of bands observed in the experimental groups subjected to *Msp I* digestion (232) was found to be lower compared to the experimental groups subjected to *Hpa II* digestion (289), considering both the bands that disappeared and those that newly appeared. In comparison to the control group, the experimental groups for *Msp I* exhibited a net increase of 157 newly created bands and a net reduction of 75 pre-existing bands. When compared to the control group for *Hpa II*, the experimental groups demonstrated a net increase of 195 newly created bands and a net loss of 94 pre-existing bands.

The observed *Msp I* polymorphism percentages exhibited a range of variability spanning from 19.5% to 50.7%. The findings of the CRED-RA test that was performed on *Msp I* indicate that the experimental group treated with 8 mg L^−1^ Ag-NPs displayed the greatest polymorphism value of 50.7%, while the experimental group treated with 2 mg L^−1^ AgNO_3_ presented the lowest value of 19.5%. According to the findings of this research, there was a rise in the polymorphism values for *Msp I* in conjunction with an increase in the concentrations of AgNO_3_, which resulted in a condition of hypermethylation. In addition, the findings of our research suggest that there was generally a rise in the polymorphism values with an increase in the Ag-NP concentrations for *Msp I*, which also resulted in a condition of hypermethylation. When looking at the polymorphism rates for *Msp I* in a more general sense, the rate of polymorphism in the experimental groups that were treated with Ag-NPs was greater than the rate of polymorphism in the experimental groups that were treated with AgNO_3_. In other words, methylation was observed to be at a higher level in the Ag-NP-treated experimental groups compared to the AgNO_3_-treated experimental groups.

The proportion of *Hpa II* polymorphism had a varied range, which spanned from 34.3% all the way up to 58.9%. The findings of the CRED-RA test that was performed on *Hpa II* indicated that the experimental group treated with 2 mg L^−1^ Ag-NPs displayed the greatest polymorphism value of 58.9%, while the experimental group treated with 6 mg L^−1^ AgNO_3_ presented the lowest value of 34.3%. It was established that the level of methylation that occurred in *Hpa II* reduced as the concentration of Ag-NPs increased in the experimental groups that were treated with Ag-NPs. On the other hand, the amount of methylation that occurred in the experimental groups that were treated with AgNO_3_ increased as the concentration of AgNO_3_ increased (Figure 6).

### 2.4. Machine Learning (ML) Analysis

In this study, we utilized the support vector machine (SVM), random forest (RF), extreme gradient boosting (XGBoost), k-nearest neighbor classifier (KNN), and Gaussian processes classifier (GP) algorithms to predict the relationship between our inputs and outputs. We compared and evaluated the performances of these models, analyzing the data generated from the experiments, including tissue culture and molecular analysis. The training dataset was employed during the model’s learning process, while the test dataset was used to assess the model’s performance. The outcomes of the machine learning models in this investigation are presented in Table 4, summarizing the findings of the study.

Metrics such as the MSE, MAPE, and MAD are the criteria that are used in the process of determining an algorithm’s overall performance. A model’s predictions are more accurate when these criteria have a lower value, since this means their values are closer to the actual values observed (Table 4). Our examination of the test performance outcomes, namely the mean squared error (MSE), mean absolute percentage error (MAPE), and mean absolute deviation (MAD), indicated a discernible trend of the XGBoost model having the highest performance, followed by GP, RF, SVM, and KNN, in the context of their CI predictions. In other words, the order of the best predictive models for CI was XGBoost < GP < RF < SVM < KNN. The XGBoost model had the greatest R^2^ value of 51.5%, indicating its superior ability to predict CI compared to the other models (Table 4).

The rankings of the most accurate prediction models in embryogenic callus (EC) was consistent based on the mean squared error (MSE) and mean absolute deviation (MAD) values but diverged when considering the mean absolute percentage error (MAPE) values. The rankings of the best prediction models for EC regarding their MSE, MAPE, and MAD values were RF < GP < SVM < KNN < XGBoost, RF < SVM < GP < KNN < XGBoost, and RF < GP < SVM < KNN < XGBoost, respectively. Overall, the random forest (RF) model had superior performance when predicting EC, as shown in all three distinct performance metrics (MSE, MAPE, and MAD). The RF model had the highest R^2^ value of 71.9%, suggesting its greater predictive capability for EC in comparison to the other models (Table 4).

The ranking of the best prediction models for forecasting regeneration efficiency (RE) was found to be consistent based on the mean squared error (MSE) and mean absolute deviation (MAD) values. However, there were discrepancies in these rankings when considering the mean absolute percentage error (MAPE) values. The rankings of the best RE prediction models regarding the MSE, MAPE, and MAD values were GP < SVM < XGBoost < RF < KNN, GP < XGBoost < SVM < RF < KNN, and GP < SVM < XGBoost < RF < KNN, respectively. Ultimately, the Gaussian processes classifier (GP) model had the superior performance when predicting RE, as shown in all three distinct performance metrics (MSE, MAPE, and MAD). The GP model had the highest R^2^ value of 52.5%, suggesting its greater predictive capability for RE in comparison to the other models (Table 4).

Additionally, the empirical data we obtained on in vitro regeneration in wheat are shown in Figure 7, together with the values that were predicted by the models. According to our results, the formula “Y = 0.596236907 X + 32.43690725” is the equation for the linear regression model that describes the relationship between the values estimated by the XGBoost algorithm, which generated the most accurate model for EC% estimation, and the actual values that were observed. The linear regression model equation “Y = 0.4973360939678 X + 34.31029044065” describes the relationship between the values estimated by the RF algorithm, which produced the most accurate model for EC estimation, and the actual observed values. The equation “Y = 0.55315933213564 X + 0.227900700516488” is the linear regression model that characterizes the association between the estimated values produced by the GP algorithm, which yielded the most perfect model for RE estimation, and the real observed values.

## 3. Discussion

Interactions between nanoparticles and plants result in a variety of morphological and physiological alterations, which are contingent upon the characteristics of the nanoparticles. The effectiveness of nanoparticles depends on factors such as their chemical composition, size, surface coverage, reactivity, and, critically, the dosage at which they are administered. Khodakovskaya et al. [57] demonstrated that nanoparticles can have both positive and negative effects on plant growth and development. It is important to note that the effectiveness of nanoparticles depends on their concentration and varies from plant to plant. The introduction of AgNO_3_ into the tissue culture medium resulted in a notable enhancement in the regenerative capacity of both monocot and dicot plants [51,58]. The precise process through which AgNO_3_ affects plants remains uncertain. However, there is limited data available that supports the involvement of this pathway in ethylene perception. AgNO_3_ has been utilized in tissue culture research to mitigate the action of ethylene, given its solubility in water and lack of phytotoxicity at effective doses [2]. In recent years, it has been demonstrated in many plants that silver nanoparticles (Ag-NPs), including AgNO_3_, have a positive effect when added to tissue culture. [10,59,60,61]. The potential beneficial impacts of Ag-NPs on in vitro parameters such as callus induction, regeneration, and micropropagation might be attributed to their ability to suppress ethylene synthesis within the growth medium. In this research, CI%, EC%, and the RE all altered based on the kind and amounts of treatment (AgNO_3_ and Ag-NPs). The highest in vitro measures, including CI rate (95.00%), EC rate (85.00%) and RE number (0.68), were seen when using the MS medium with a concentration of 2 mg L^−1^ of silver nanoparticles (Ag-NPs). The lowest values of EC% rate, CI% rate and RE number were obtained from MS medium with 4 mg L^−1^ AgNO_3_. The gaseous ingredients inside the tissue culture system, with a particular emphasis on ethylene, have a significant influence on the growth and developmental processes of plants [2,6]. According to the literature, the presence of ethylene has been shown to hinder the process of somatic embryogenesis and the production of shoot primordia in callus cultures [62]. The potential benefits of AgNO_3_, acting as a competitor for the binding site of ethylene, on plant tissue culture have been documented in several plant species. [63,64]. The use of silver nitrate, particularly in cereals, has been seen to have a positive effect on somatic embryogenesis. In buffalograss, the treatment of AgNO_3_ was shown to considerably enhance callus induction frequency as well as growth, according to the findings of Fei et al. [65]. The callus induction frequency reached its maximum value of 79.9% when exposed to a concentration of 10 mg L^−1^ AgNO_3_. It was observed that the application of AgNO_3_ had a positive effect on the initiation of callus formation in *Paspalum scrobiculatum* and *Eleusine coracana*. Based on their research results, it was observed that the use of MS medium enriched with AgNO_3_ resulted in the development of embryogenic and friable callus in both species [66]. According to the literature, the addition of AgNO_3_ has been shown to increase the development of embryogenic callus in bread wheat [62]. So far, there have been limited studies conducted on the effects of Ag-NPs on various plant species, including *Phaseolus vulgaris* [67], *Eruca sativa* [68], *Lolium multiflorum* [69], *Sorghum bicolor* [70], *Arabidopsis thaliana* [71], *Allium cepa* [5], and *Oryza sativa* [72]. Positive response of Ag-NPs added to tissue culture has been reported in many plant species [61,73,74]. Various studies have shown the beneficial impacts of NPs on the processes of callus induction, shoot regeneration, and growth. Sarmast et al. [75] reported significant differences in the mean number of shoots per explants, mean length of shoots per explants and percentage of explants producing shoots between explants grown in MS medium supplemented with 60 µg ml^−1^ Ag-NPs compared to controls. Silver nanoparticles (30 mg L^−1^) added to a culture medium have been shown to increase biomass production in cell suspension cultures of *Arabidopsis thaliana* [60]. Karimi and Mohsenzadeh [76] documented that wheat plants exhibited a notable and significant decline in growth when subjected to two concentrations of AgNO_3_ and Ag-NPs treatments (10 mg L^−1^, 100 mg L^−1^). Based on our findings, higher doses of AgNO_3_ administration resulted in a general rise in the induction of embryogenic callus in bread wheat. On the contrary, concentrations increasing of Ag-NPs treatment led to a reduction in CI rate, EC rate and RE number. When Ag-NPs were applied at greater doses, negative effects on plant development were also seen in *Saccharum* spp. [77], *Vanilla planifolia* [78], and *Prunus armeniaca* [22]. According to Castro-González et al. [79], elevated concentrations of this compound may have a negative impact on the cellular processes, including cell divisions. According to Timoteo et al. [80], the application of Ag-NPs at low concentrations resulted in an increase in both the fresh and dry weights of *Physalis peruviana* seedlings. However, greater concentrations of Ag-NPs were shown to inhibit shoot development and root formation. On the other hand, the exposure to silver nanoparticles (Ag-NPs) was shown to impede the development and biomass accumulation of *Spirodela polyrhiza*, as reported by Jiang et al. [81]. To a certain degree, the plant has the potential to endure an elevation in the concentration of metallic particles, and it may even aid development to a limited level. However, it has been shown that elevated levels of metal ions have detrimental effects on the development of plants [82]. The potential impacts of high concentrations of Ag-NPs on plants might vary depending on factors such as the specific kind of plant, its genetic structure, morphology, biochemistry, and physical properties.

Although its underlying mechanisms are not fully understood, the properties of tissue culture are thought to be a potential source of genomic instability [83]. The advancement of modern methodologies has revealed that plant tissue culture is subject to epigenetic modifications, such as histone methylation/demethylation [84], DNA methylation levels [85], alterations in gene expression [86], and the involvement of various small RNAs [87]. In the present study, RAPD techniques were employed to assess genetic variability. The results presented in Table 2 indicate the existence of genetic variability among wheat plants exposed to different concentrations of Ag-NPs and AgNO_3_, as compared to the control group. This finding is consistent with that of Abdel-Azeem and Elsayed [88], whom studied the effects of silver engineered nanoparticles on *Vicia faba* seedlings and discovered that the particles altered mitotic indices and chromosomal morphology, causing metaphase and anaphase chromosome lag, chromosome fragmentation and bridging, chromosome stickiness, and micronuclei. Ewais et al. [89] investigated genetic variation in DNA samples treated with Ag-NPs and a control treatment of *Solanum nigrum* using RAPD techniques and reported that there were different levels of genetic variation, which is consistent with our study according to our RAPD analysis results.

It is expected that regenerants produced by somatic embryogenesis will be phenotypically and genetically indistinguishable from their parent plant. However, this cannot be assumed as a rule for all cases. Regenerants can show phenotypic, cytological, and genetic alterations [90]. The factors contributing to variance in plant tissue grown in vitro are diverse, including the components of the culture media, the duration of plant tissue cultivation, and the environmental conditions [91,92]. For instance, among the components of the culture medium, micronutrients like copper or silver appear to influence the process of somatic embryogenesis, as has recently been revealed in research on genetic and epigenetic modifications [39]. During the process of cell differentiation and dedifferentiation in plant regeneration systems, epigenetic variation has been observed both during and after the cells were subjected to the conditions of in vitro culture [93]. DNA sequence variation is the essential evolutionary mechanism generating phenotypic variances [94], whereas DNA methylation changes in may impact gene expression and thus may conceivably contribute to trait differences that may be passed on to future generations. DNA methylation plays a crucial role in proper cellular development cellular differentiation [95]. The effects of various nanomaterials and nanoparticles on DNA methylation have been studied extensively in recent years [30,32]. In the present research, we assessed the average proportion of DNA methylation-induced polymorphism across all the experimental groups using the CRED-RA test. In our study, the *Msp I* experimental groups showed a net increase of 157 newly formed bands and a net decrease of 75 pre-existing bands when compared to the control group and, the experimental groups for *Hpa II* showed a net increase of 195 newly produced bands and a net loss of 94 pre-existing bands when compared to the control group. A wide variation was seen in the percentages of *Msp I* polymorphism, from 19.5% to 50.7%. The CRED-RA analysis conducted on *Msp I* determined that the experimental group treated with 8 mg L^−1^ Ag-NPs provided the highest polymorphism value (50.7%), whereas the experimental group treated with 2 mg L^−1^ AgNO_3_ presented the lowest value (19.5%). According to CRED-RA analysis, the rate of polymorphism in experimental groups treated with Ag-NPs was higher than the rate of polymorphism in experimental groups treated with AgNO_3_. In other words, methylation levels appeared to be higher in Ag-NPs treated experimental groups when compared to AgNO_3_-treated experimental groups. The process of modifying the chromatin structure via DNA and histone changes, known as epigenetic modulation, plays a crucial role in regulating gene expression at the transcriptional level in response to environmental stimuli. DNA methylation plays a crucial role in shaping chromatin structure and regulating gene expression by influencing the accessibility of genes to the transcriptional machinery, a well-recognized phenomenon in the field. In both AgNO_3_ application and Ag-NPs application, an increase in hyper methylation was observed with an increase in concentration, and the results are similar to previous studies [32,33]. It is noteworthy to emphasize that DNA cytosine methylation plays a crucial role as a checkpoint control mechanism in regulating cellular transcription programs. The present study suggests that AgNO_3_ and Ag-NPs may induce epigenetic modifications in the genome, resulting in the modulation of growth, physiology, anatomy, tissue differentiation, and metabolism.

In the scope of this study, output variables (CI, EC, and RE) were targeted by making use of input components and molecular values, and estimate models were assessed by means of ML algorithms. The ML algorithms are very suitable for analyzing and validating projected output variables due to their ability to evaluate the input parameters associated with the desired results [47,48,49,50]. In recent times, there has been a growing use of machine learning models in the field of in vitro regeneration research for the purpose of data validation. These models have been utilized in a diverse range of studies, each with its own specific objectives and goals [47,48,96]. To the best of our knowledge, this research endeavor marks the first instance of employing ML techniques to analyze the efficacy of AgNO_3_ and Ag-NPs in vitro regeneration, considering their concentrations and molecular effects as input factors. The results of our research indicated that the XGBoost, RF, and GP algorithms, in that order, are the ones that perform the best when it comes to the estimation of CI, EC, and RE, which are the in vitro regeneration parameters of wheat. In parallel, in a recent study conducted by Kirtis et al. [55], it was shown that XGBoost had the highest predictive performance when compared to three other machine learning algorithms (support vector regression, gaussian process regression, and random forest) in the context of modeling the in vitro regeneration of chickpea (*Cicer arietinum* L.). Similarly, RF (random forest) and XGBoost (extreme gradient boosting) have emerged as suitable candidates for predicting shot counts and shoot length in Alternanthera reineckii [48]. Aasim et al. [50] demonstrated the capability of Cannabis sativa to generate an optimal model using the RF algorithm for the prediction and validation of germination and growth indices. According to the findings of our research, the use of a mixture of XGBoost, RF, and GP algorithms demonstrates superior predictive capabilities for in vitro parameters. This amalgamation of algorithms has promised as a valuable tool for improving and forecasting outcomes in in vitro culture systems.

## 4. Materials and Methods

### 4.1. Synthesis of Silver Nanoparticles (Ag-NPs)

In this study, silver nanoparticles (Ag-NPs) were synthesized using *Eruca vesicaria* plant extracts. Briefly, after washing 2–3 times (with tap water and distilled water), the *Eruca vesicaria* plant leaves were cut into small pieces and transferred into a glass beaker containing 200 mL of distilled water. After the process of heating at 75 °C for 10 min and boiling for 5 min, it was cooled to room temperature and filtered using Whatman filter paper no.1. Then, the *Eruca vesicaria* extract was stored at 4 °C for Ag-NP synthesis. The nanoparticles were characterized using a UV-Vis spectrophotometer (Shimadzu, UV-3600 Plus, Kyoto, Japan), transmission electron microscopy (TEM) (Hitachi HighTech HT7700, Tokyo, Japan), and X-ray diffractometer (XRD) (PANalytical, Empyrean, the Netherlands), as in previous study [97]. Ag-NPs were obtained by adding 10 mL of plant extract to a solution of AgNO_3_ (Sigma-Aldrich, cat. no: 7761-88-8, St. Louis, MO, USA) (500 mL, 1 mM). A color change (from brown to red) was observed, followed by centrifugation, washing, and drying processes at 75 °C. One previous study [97] has also presented the results of a characterization study on Ag-NPs. These nanoparticles had sizes ranging from 5 to 20 nm and exhibited spherical, triangular, and cubic shapes. The study found that Ag-NPs showed absorption peaks at 250 and 450 nm. Ag-NPs were also found to have a face-centered cubic crystal structure.

### 4.2. Plant Material

For the plant material in this investigation, a Kırik wheat (*Triticum aestivum* L.) cultivar was used. The seeds, obtained from the Faculty of Agriculture at Ataturk University, were washed with regular water and surface-sterilized with 70% ethanol for five minutes, and then subjected to treatment for twenty-five minutes with a solution containing one percent sodium hypochlorite and a few drops of Tween-20, with steady stirring. Finally, the seeds were washed three times with sterile distilled water. The seeds were stored in a controlled environment at a temperature of 4 °C and subjected to a period of darkness lasting 16–18 h. To induce callus formation, mature wheat embryo seeds were cultured on Murashige and Skoog (MS) [98] medium.

### 4.3. In Vitro Conditions

Mature seeds underwent surface sterilization initially by being washed in 70% ethanol for 5 min, followed by rinsing with sterile water. Subsequently, the seeds were treated with a 10% solution of sodium hypochlorite containing two drops of Tween-20 for 20 min. Afterward, the seeds were rinsed three times with sterile water and then incubated at 4 ºC for 24 h in sterile distilled water. Imbibed seeds were prepared according to the method described by Türkoğlu [99].

The callus induction medium used in this study consisted of a combination of MS, 12 mg L^−1^ dicamba, 0.5 mg L^−1^ IAA (Indole-3-Acetic acid), 20 g L^−1^ sucrose, 2 g L^−1^ phytagel, and 1.95 g L^−1^ MES (Sigma-Aldrich, St. Louis, MO, USA). The medium also included different treatments, namely silver nitrate (AgNO_3_) and silver nanoparticles (Ag-NPs), at concentrations of 0, 2, 4, 6, and 8 mg L^−1^. The pH of each medium was brought to its ultimate value of 5.8 using 1 N NaOH. The pH of the media was adjusted to 5.8 using sodium hydroxide (NaOH) before autoclaving at 121 ℃ and 105 kPa for 20 min. Vitamins and plant growth regulators went through a process of filter-based sterilizing. The Petri dishes were incubated in darkness at 25 ± 1 °C for 30 days to induce callus formation from endosperm-supported mature embryos. After, callus induction ratio (CI%), embryogenic callus ratio (EC%), and regeneration efficiency (RE) were determined [100,101]. After 30 days of culturing, the calluses were placed into Petri dishes containing MS (Murashige and Skoog) medium supplemented with 0.5 mg L^−1^ TDZ (N-Phenyl-N′-1,2,3-thiadiazol-5-ylurea, Thidiazuron), 20 g L^−1^ sucrose, and 7 g L^−1^ agar, which is the medium needed for embryonic callus development and plant regeneration. Solidification of the media, adjustment of the pH, and sterilization were performed in accordance with the instructions outlined for the callus induction media. After that, they were cultivated for 30 days at 25 °C with a photoperiod of 16 h of light (62 mol m^−2^ s^−1^) and 8 h of darkness. EC (%) was calculated as the number of embryonic callus es/number of explants × 100, and RE was calculated as the number of regenerated plants/numbers of embryonic calluses. The plants were transferred into magenta boxes containing the same regeneration material and kept at the same plant regeneration settings once they had grown to a height of 10–12 cm [100,101]. The present experiments were performed using a completely randomized factorial design (two different Ag substances (AgNO_3_ and Ag-NPs) × five different concentrations (0, 2, 4, 6, and 8 mg L^−1^), with four replications and ten explants per replicate. Each Petri dish was considered as one experimental unit, and ten mature embryos were cultured on each dish. The statistical software package GenStat v. 23, developed by VSN, was utilized to perform a two-way analysis of variance (ANOVA). The treatment means and interactions were compared using Duncan’s multiple range test.

### 4.4. Molecular Assays

#### 4.4.1. Isolation of Genomic DNA

Genomic DNA was isolated from the cultured embryogenic calluses. Briefly, DNA samples were prepared using the CTAB method from the study by Zeinalzadehtabrizi et al. [102]. After that, the DNA was preserved for later use by being stored at a temperature of −20 °C. A nanodrop spectrophotometer (Thermo Fisher Scientific, Waltham, MA, USA) and electrophoresis on a 0.8% agarose gel were used to determine the concentration of the genomic DNA as well as its level of purity.

#### 4.4.2. RAPD and CRED-RA PCR Assays

A total of fifteen random amplified polymorphic DNA (RAPD) primers consisting of 15 nucleotides were used for the purpose of analysis. Only eight primers were found to successfully amplify polymorphic bands and were then used in polymerase chain reaction (PCR) for RAPD and CRED-RA analyses (Table 5). To conduct RAPD analysis, a polymerase chain reaction was executed with a total volume of 20 µL. This volume included of 10× PCR buffer, 10 mM dNTP mixture, 25 mM MgCl_2_, ddH_2_O, 10 pmol of random primer, 1 U of Taq DNA polymerase (Thermo Fisher Scientific, Waltham, MA, USA), and 50 ng µL^−^^1^ DNA sample. Afterward, the prepared cocktail in the tubes was placed in a thermocycler (Senso Quest Lab cycler, manufactured in Gottingen, Germany) for amplification. The first stage of the PCR process consisted of an initial denaturation phase, which was performed at a temperature of 95 °C for five minutes. After that, a total of forty cycles were performed, each of which consisted of denaturation for one minute at 95 °C, annealing for one minute at the ideal annealing temperature of the marker employed, extension for two minutes at 72 °C, and final extension for ten minutes at 72 °C.

To conduct a CRED-RA analysis, a quantity of 1 µg of DNA sample was individually digested at 37 °C for 2 h using 1 µL of *Hpa II* and 1 µL of *Msp I*, as directed by the manufacturer (Thermo Scientific, Waltham, USA). The DNA samples that were subjected to digest were then used as templates in the polymerase chain reaction (PCR) process. The process for amplification was carried out by employing the primers that are presented in Table 5. The methodological stages of the PCR were like those used in the RAPD investigation. The electrophoresis approach was used to differentiate the RAPD and CRED-RA PCR products based on their base size [38,103].

#### 4.4.3. Genetics Analysis

To analyze the RAPD and CRED-RA banding patterns, a TotalLab TL120 software (Nonlinear Dynamics Ltd., Newcastle, UK) was used. When compared to the control, the RAPD profiles revealed polymorphism when a band that was predicted to be present did not exist and a novel band did appear. Changes in the average polymorphism were calculated for each treatment group (changing concentrations of AgNO_3_ and Ag-NPs), and the results were reported as a percentage in comparison to the value obtained from the control (which was set to 100%) [45,46,104,105]. A quantitative metric known as genomic template stability (GTS%) was estimated for RAPD using the formula GTS = (1 − a/n) × 100. In this calculation, *a* represents the mean number of polymorphic bands in each treated template, and *n* represents the total number of bands in the control [38,103]. With the use of the formula Polymorphism = a/n × 100 [45,46,104,105], we were able to calculate the average polymorphism values (in percent) for each concentration.

### 4.5. Modeling Using Machine Learning Algorithms

In order to train the model and estimate the output variables of tissue culture parameters [CI, EC, and RE], the following ML methods were applied: support vector machines (SVM) [106], random forest (RF) [107], extreme gradient boosting (XGBoost) [108], k-nearest neighbors classifier (KNN) [109], and gaussian processes classifier (GP) [110]. In the data set, the inputs that were used comprised two separate experimental design treatments (AgNO_3_ and Ag-NPs), as well as the concentration of treatments (0, 2, 4, 6, and 8 mg L^−1^). Furthermore, the data set included the inclusion of GTS rates, *Msp* I polymorphism, and *Hpa* II polymorphism in wheat, which were influenced by the experimental groups under investigation. As a result, the estimations on the observed variables (CI, EC, and RE) were derived based on the influence of both the treatments applied in MS medium and the resulting variations in gDNA. The wheat dataset was partitioned into two separate sets of data, referred to as the training set and the test set, with a ratio of 70% to the training set and 30% to the test set, respectively. To determine how well the models were performing, the leave-one-out cross-validation (LOO-CV) method was used [101]. To conduct an accurate analysis of the performance of each model, a total of four separate performance indicators, namely R^2^, MSE, MAPE, and MAD, were used.

The degree of correlation that exists between the model and the dependent variables was evaluated using a calculation of the coefficient (R^2^), which can be found in Equation (1). The mean squared error (MSE) is a statistical metric employed to assess the accuracy of a regression model by measuring the mean squared difference between the actual and predicted values. Calculating the mean squared error (MSE) involves finding the average of the squared deviations that occur between the values that were seen and the values that were anticipated using Equation (2). The mean absolute percentage error (MAPE) is a typical error metric that is used in regression analysis. It is a statistical measure that determines the average of the absolute differences that exist between the values that were predicted and those that really occurred for each data point.

These differences are then reported as a percentage (%) of the value that occurred. The mean absolute percentage error, also known as MAPE, has the virtue of scale independence, which means that changes may be expressed as percentages. This makes it possible to evaluate datasets that are not identical to one another. This capability enables comparisons to be made across many different datasets (Equation (3)). The mean absolute deviation (MAD) is calculated by first determining the absolute difference between the actual value and the anticipated value for each data point, and then determining the average of these absolute differences. Since it uses absolute values of differences in its calculation (Equation (4)), the mean absolute deviation (MAD) measure is sensitive to the presence of substantial outliers [47,48,49,50,54,96,111,112].
(1)$$R^{2}=1-\left(\frac{\sum_{i=1}^{n}{\left(y_{i}-y_{ip}\right)}^{2}}{\sum_{i=1}^{n}{\left(y_{i}-\bar{y}\right)}^{2}}\right)$$
(2)$$MSE=\sqrt{\frac{1}{n}\sum_{i=1}^{n}{\left(y_{i}-y_{ip}\right)}^{2}}$$
(3)$$MAPE=\frac{1}{n}\sum_{i=1}^{n}\left|\frac{y_{i}-y_{ip}}{y_{i}}\right|\times 100$$
(4)$$MAD=\frac{1}{n}\sum_{i=1}^{n}\left|y_{i}-y_{ip}\right|$$
where n is the total number of samples used for training and testing, y_i_ is the actual value that was measured, *y*_ip_ is the value that was predicted, and $\bar{y}$ is the mean of the measured values. The ML methods and performance metrics were both computed with the help of the R software [113,114].

## 5. Conclusions

To facilitate the progress of plant tissue culture techniques, it is essential that regenerated plantlets exhibit consistent genetic and morphological stability. The findings of this study provide evidence that the inclusion of AgNO_3_ relative to Ag-NPs in the growth medium had a substantial positive impact on the CI%, EC%, and RE values of *Triticum aestivum* L. These results offer new perspectives on the detrimental or beneficial effects of Ag-NPs as a potent antagonist in growth media. Potential epigenetic changes brought on by AgNO_3_ and Ag-NPs that affect a variety of in vitro characteristics might have a negative influence on wheat in vitro regeneration. Conducting further research is recommended to further explore the evaluation of molecular and biological processes triggered by nanoparticles, as well as to gain a deeper understanding of many aspects of nanoparticle-induced toxicity in plants. Additionally, the estimation of variables and the creation of models in in vitro processes with ML algorithms have not been sufficiently investigated yet. As this study’s results indicate, a computational method that combines XGBoost, RF, and GP techniques may be a promising strategy for predicting and improving the in vitro characteristics of wheat.

## Figures and Tables

**Figure 1 plants-12-04151-f001:**
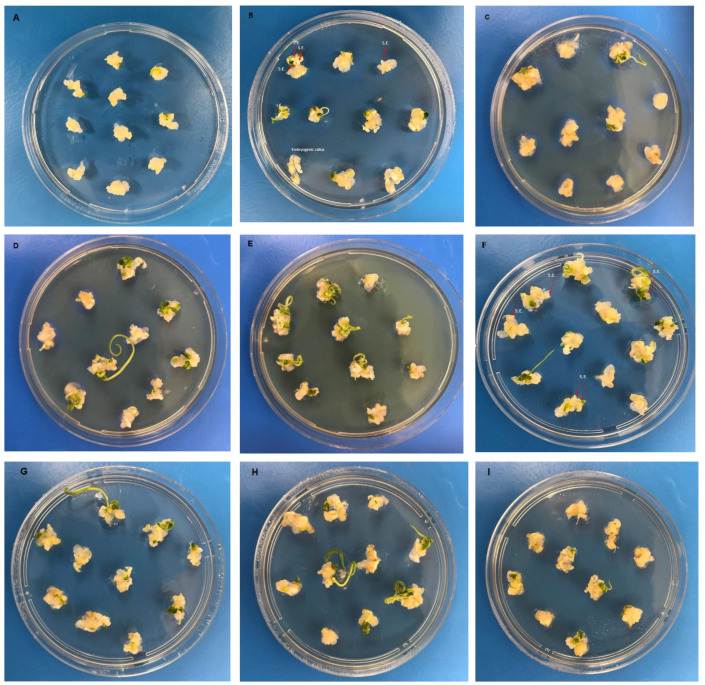
Embryogenic callus induction of wheat (*Triticum aestivum* L.) on MS medium supplemented with different concentrations of AgNO_3_ and Ag-NPs. (**A**): Control (without AgNO_3_ and Ag-NPs); (**B**): 2 mg L^−1^ AgNO_3_ treatment; (**C**): 4 mg L^−1^ AgNO_3_ treatment; (**D**): 6 mg L^−1^ AgNO_3_ treatment; (**E**): 8 mg L^−1^ AgNO_3_ treatment; (**F**): 2 mg L^−1^ AgNO_3_-NP treatment; (**G**): 4 mg L^−1^ AgNO_3_-NP treatment; (**H**): 6 mg L^−1^ AgNO_3_-NP treatment; (**I**): 8 mg L^−1^ AgNO_3_-NP treatment. The red arrows indicate the presence of somatic embryos (S.E.) in (**B**,**F**).

**Figure 2 plants-12-04151-f002:**
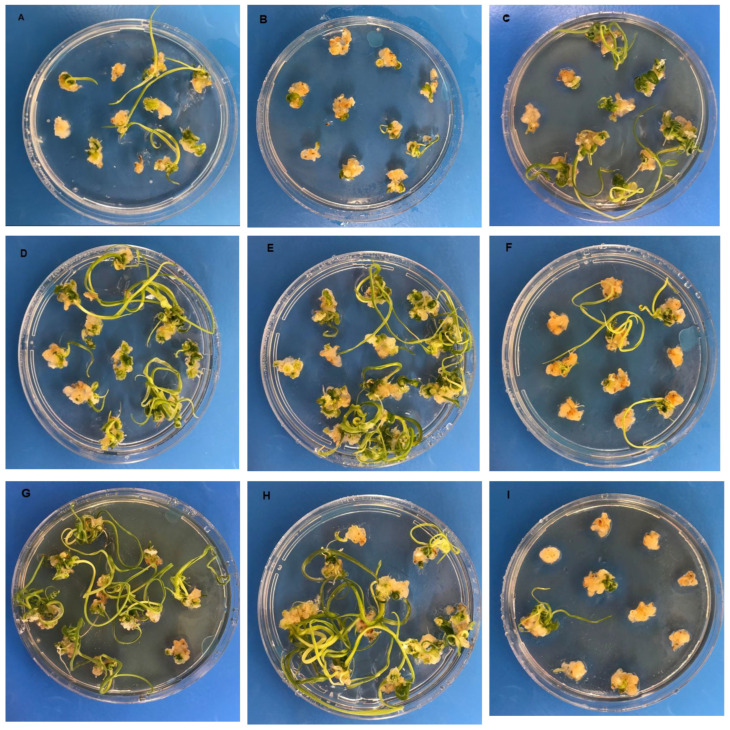
Regeneration plants occurring from the embryonic calluses in wheat (*Triticum aestivum* L.) on MS medium supplemented with different concentrations of AgNO_3_ and Ag-NPs. (**A**): Control (without AgNO_3_ and Ag-NPs); (**B**): 2 mg L^−1^ AgNO_3_ treatment; (**C**): 4 mg L^−1^ AgNO_3_ treatment; (**D**): 6 mg L^−1^ AgNO_3_ treatment; (**E**): 8 mg L^−1^ AgNO_3_ treatment; (**F**): 2 mg L^−1^ AgNO_3_-NP treatment; (**G**): 4 mg L^−1^ AgNO_3_-NP treatment; (**H**): 6 mg L^−1^ AgNO_3_-NP treatment; (**I**): 8 mg L^−1^ AgNO_3_-NP treatment.

**Figure 3 plants-12-04151-f003:**
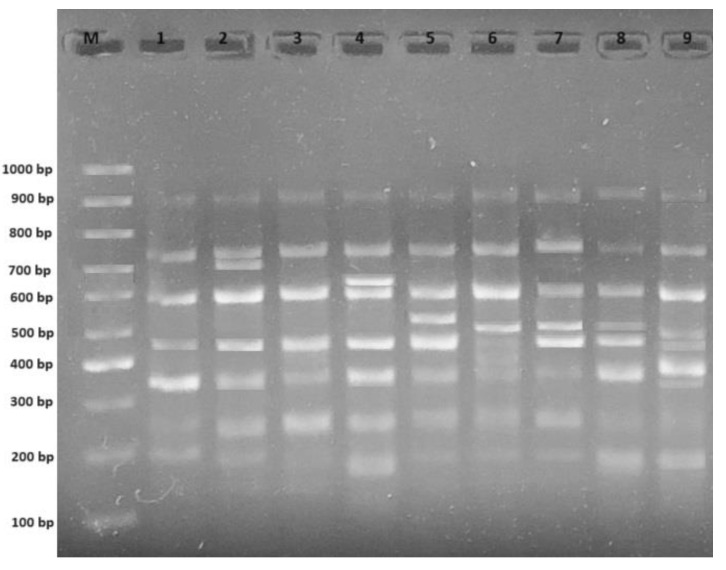
RAPD profiles of various experimental groups with OPW-06 primer. M: 100–1000 bp DNA ladder; 1: control; 2: 2 mg L^−1^ AgNO_3_ treatment; 3: 4 mg L^−1^ AgNO_3_ treatment; 4: 6 mg L^−1^ AgNO_3_ treatment; 5: 8 mg L^−1^ AgNO_3_ treatment; 6: 2 mg L^−1^ AgNO_3_-NP treatment; 7: 4 mg L^−1^ AgNO_3_-NP treatment; 8: 6 mg L^−1^ AgNO_3_-NP treatment; 9: 8 mg L^−1^ AgNO_3_-NP treatment.

**Figure 4 plants-12-04151-f004:**
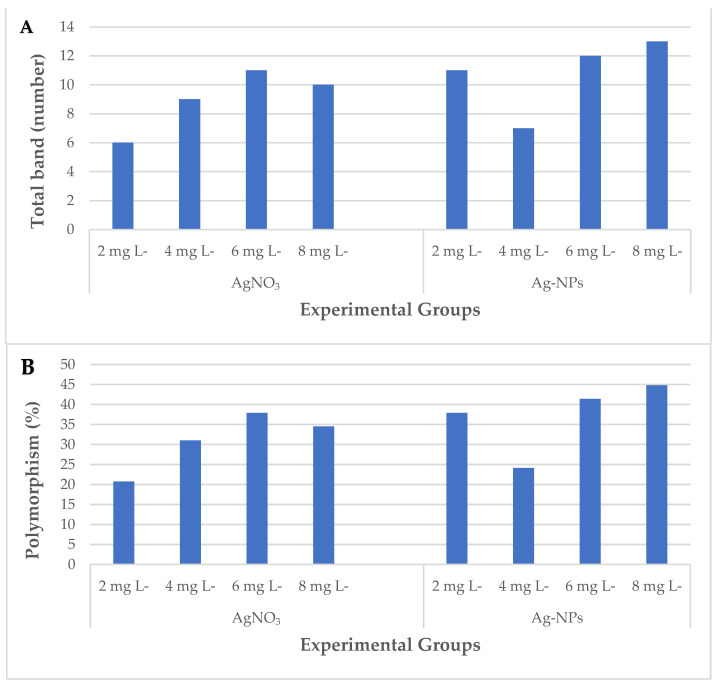
DNA methylation changes in the wheat exposed to AgNO_3_ and Ag-NPs. (**A**) Total band; (**B**) polymorphism; (**C**) GTS value, as estimated using different experimental groups.

**Figure 5 plants-12-04151-f005:**
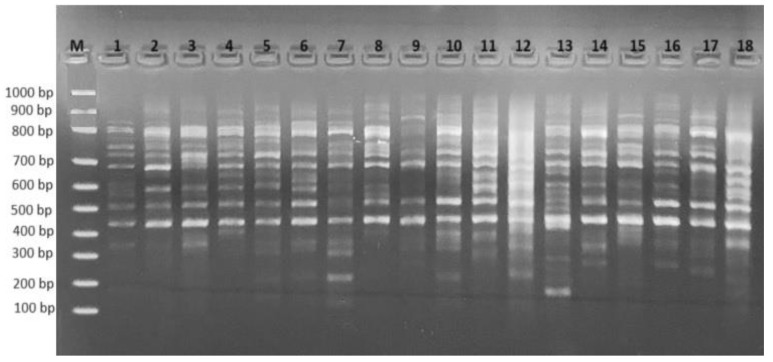
CRED-RA profiles of various experimental groups with OPW-06 primer. M: 100–1000 bp DNA ladder; 1: control *Hpa II*; 2: control *Msp I*; 3: 2 mg L^−1^ AgNO_3_-treated *Hpa II*; 4: 2 mg L^−1^ AgNO_3_-treated *Msp I*; 5: 4 mg L^−1^ AgNO_3_-treated *Hpa II*; 6: 4 mg L^−1^ AgNO_3_-treated *Msp I*; 7: 6 mg L^−1^ AgNO_3_-treated *Hpa II*; 8: 6 mg L^−1^ AgNO_3_-treated *Msp I*; 9: 8 mg L^−1^ AgNO_3_-treated *Hpa II*; 10: 8 mg L^−1^ AgNO_3_-treated *Msp I*; 11: 2 mg L^−1^ AgNO_3_-NP-treated *Hpa II*; 12: 2 mg L^−1^ AgNO_3_-NP-treated *Msp I*; 13: 4 mg L^−1^ AgNO_3_-treated *Hpa II*; 14: 4 mg L^−1^ AgNO_3_-treated *Msp I*; 15: 6 mg L^−1^ AgNO_3_-treated *Hpa II*; 16: 6 mg L^−1^ AgNO_3_-treated *Msp I*; 17: 8 mg L^−1^ AgNO_3_-treated *Hpa II*; 18: 8 mg L^−1^ AgNO_3_-treated *Msp I*.

**Figure 6 plants-12-04151-f006:**
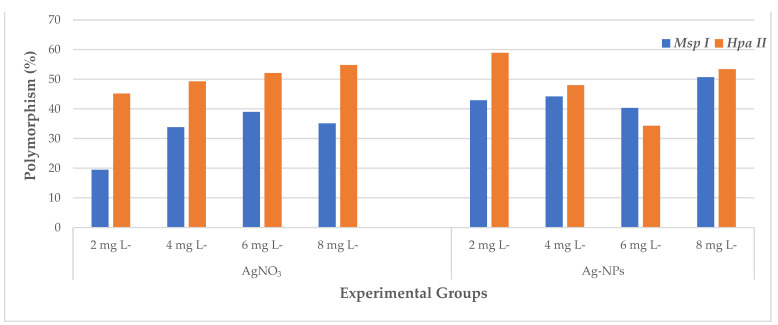
The effects of AgNO_3_ and Ag-NPs on polymorphism percentages in different experimental groups of wheat.

**Figure 7 plants-12-04151-f007:**
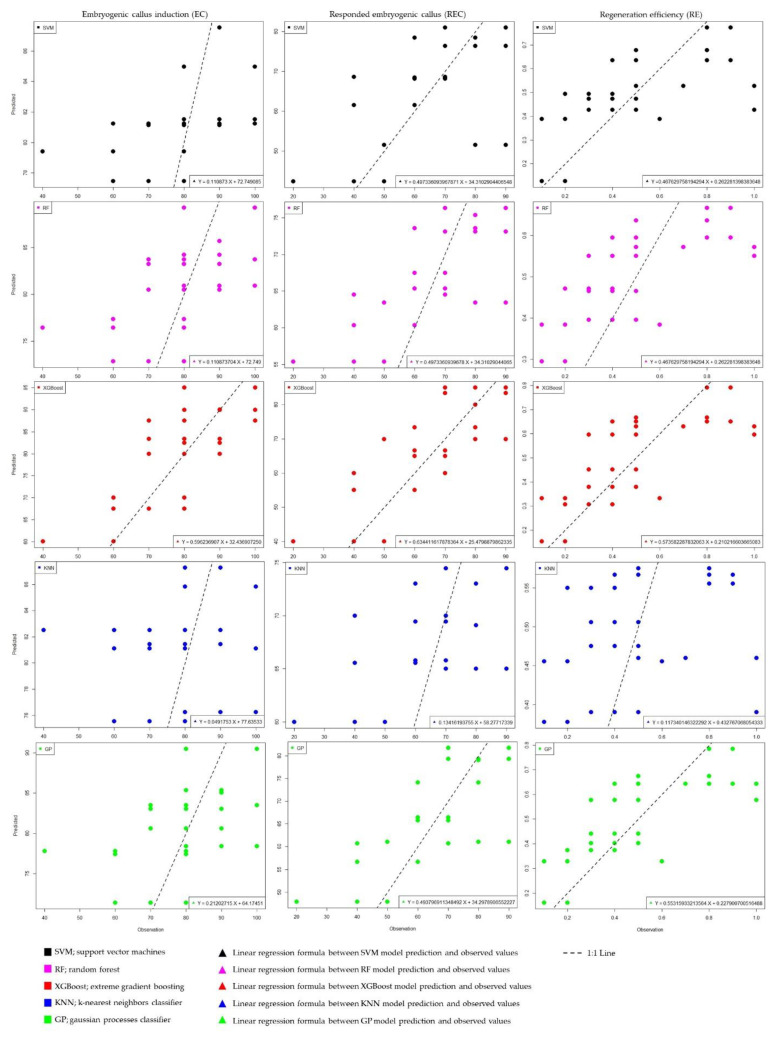
The observed actual values of the in vitro regeneration of wheat and the values predicted by the ML models.

**Table 1 plants-12-04151-t001:** Average values and analyses of variance of AgNO_3_ and Ag-NPs at different doses examined in wheat plant.

Treatment	Concentration (mg L^−1^)	CI (%) ^1^	EC (%)	RE (Number)
AgNO_3_	0	80.00 ^abc2^	65.00 ^bc^	0.43 ^bcd^
AgNO_3_	2	82.50 ^abc^	70.00 ^abc^	0.20 ^d^
AgNO_3_	4	60.00 ^d^	37.50 ^d^	0.13 ^d^
AgNO_3_	6	90.00 ^ab^	80.00 ^ab^	0.73 ^ab^
AgNO_3_	8	90.00 ^ab^	80.00 ^ab^	0.83 ^a^
	Means	80.50	66.50	0.46
Ag-NPs	0	80.00 ^abc^	65.00 ^bc^	0.43 ^bcd^
Ag-NPs	2	95.00 ^a^	85.00 ^a^	0.68 ^ab^
Ag-NPs	4	87.50 ^ab^	67.50 ^abc^	0.55 ^abc^
Ag-NPs	6	75.00 ^bcd^	62.50 ^bc^	0.65 ^ab^
Ag-NPs	8	67.50 ^cd^	55.00 ^c^	0.33 ^cd^
	Means	81.00	67.00	0.53
Mean concentration	0	80.00	65.00 ^a^	0.43 ^bc^
	2	88.75	77.50 ^a^	0.44 ^bc^
	4	73.75	52.50 ^b^	0.34 ^c^
	6	82.50	71.25 ^a^	0.69 ^a^
	8	78.75	67.50 ^a^	0.58 ^ab^
Mean square of treatment (T)		2.50 ^ns^	2.50 ^ns^	0.04 ^ns^
Mean square of concentration (C)		241.25 ^ns^	685.00 **	0.15 *
Mean square of T × C		821.25 ***	1027.50 ***	0.32 ***

^1^ CI: callus induction (%); EC: embryogenic callus induction (%); and RE: regeneration efficiency (number). ^2^ Letters of the same notation indicate important items. ns: non-significant at *p* ≥ 0.05; ** significant at *p* ≤ 0.01; * significant at *p* ≤ 0.05; *** significant at *p* ≤ 0.001.

**Table 2 plants-12-04151-t002:** Molecular sizes (bp) of appearing/disappearing bands in the RAPD profiles of wheat treated with varying concentrations of AgNO_3_ and Ag-NPs.

Primers	± ^1^	Control ^2^	Experimental Groups							
			AgNO_3_				Ag-NPs			
			2 mg L^−1^	4 mg L^−1^	6 mg L^−1^	8 mg L^−1^	2 mg L^−1^	4 mg L^−1^	6 mg L^−1^	8 mg L^−1^
OPA 4	+	4	-	-	-	-	-	-	-	679; 560; 542; 368
	-		458	-	-	-	458	-	-	-
OPH 17	+	1	-	-	375; 216	628; 466; 207	-	-	588	418
	-		-	-	295	295	-	-	295	295
OPH 18	+	2	574	267	634	567	629; 226	-	604; 571; 285	365
	-		467	500	-	467	-	-	-	-
OPW 4	+	6	549; 412	481; 457	436	481	478; 457	-	471	488
	-		-	-	-	-	297	812; 297	812	-
OPW 6	+	7	706	-	618	531	502; 367	615; 502	500	413; 331
	-		-	-	-	-	-	-	-	-
OPW 11	+	3	-	-	511; 469; 400	-	-	-	478	-
	-		-	-	-	-	-	418	-	-
OPW 17	+	2	-	629; 312; 208; 156	-	-	-	-	613	788
	-		-	-	400	400	400	400	400	-
OPW 20	+	4	-	500	162	408	412; 236	352	329	527; 111
	-		-	-	-	-	-	-	-	

^1^ appearance of a new band (+), and disappearance of a normal band (-); ^2^ without AgNO_3_ and Ag-NPs, respectively.

**Table 3 plants-12-04151-t003:** Molecular size of bands and polymorphism percentages according to the CRED-RA analysis.

Primers	M*/H* ^1^		± ^2^	Control ^3^	Experimental Groups							
					AgNO_3_				Ag-NPs			
					2 mg L^−1^	4 mg L^−1^	6 mg L^−1^	8 mg L^−1^	2 mg L^−1^	4 mg L^−1^	6 mgL^−1^	8 mg L^−1^
OPA 4	M		+	9	-	647; 585	576; 515; 402; 274	616; 564; 526	668; 600; 547	708; 622; 544; 508	654; 576; 505	691; 641; 564; 528; 512
			-		-	827	-	871; 827	-	-	-	827
	H		+	7	951; 330	907; 751; 600; 498; 433; 406; 289; 256; 227	1000; 916; 792; 550; 489; 433; 317; 294; 242	990; 541; 498; 472; 323; 160	1018; 907; 768; 641; 560; 489; 412; 298	907; 852; 792; 708; 553; 538; 519; 474; 416; 360; 330; 307	-	900; 871; 725; 491; 472; 319
			-		-	-	-	-	-	-	-	-
OPH 18	M		+	15	-	-	-	-	1427; 800; 514	1427	-	1472
			-		1309; 337; 286	1145; 708; 582; 337; 286	1145; 1063; 882; 708; 666; 286	708; 337; 286	286	1145; 666; 286	708; 337; 286	420; 286
	H		+	11	-	-	471	1018; 832; 492; 359	467	380; 328	1145; 1000	-
			-		1336; 1109; 775; 683; 627	1336; 891; 775; 683	1336; 1181; 891	1181	775	1109	-	627
OPW 4	M		+	11	-	731; 549	-	-	600; 434; 306	445	543; 310	800; 434
			-		362; 259; 205	362; 205	362; 205	205	305	-	841	-
	H		+	13	439	-	-	777; 434	296	-	-	-
			-		900; 600; 327; 149	900; 629; 600; 516; 362; 149	900; 600	900; 629; 600;	900; 629; 600;	831; 600; 516	900; 600; 516	900; 831; 600; 570; 391
OPW 5	M	+		6	491	813;497; 338	1318; 864; 654 502; 338; 323	826; 494; 354;331; 305	852; 747; 578; 494; 406; 313	888; 760; 578; 482; 360; 338	864; 711; 639;570; 488; 403; 333; 297; 206	930; 839; 722; 632; 488 450; 346;320; 292
		-			-	-	-	-	-	-	-	-
	H	+		12	826	900; 760; 679; 600; 524; 510; 264	983; 930; 760; 604	826; 502	800; 711	921; 734; 600; 294	864	888; 773; 513; 292
		-			639; 482; 394; 373; 352; 333	-	415; 394; 333; 252; 200	373	394; 373	394	532; 415; 333; 200	415; 394
OPW 6	M	+		11	829; 296	854; 749; 466; 145	800; 154	966; 866; 715	955; 866; 766	955; 866; 700; 641; 449; 352; 228;179	900; 800; 700; 312	900; 800; 715; 191
		-			-	-	-	485; 429	-	-	485	-
	H	+		12	605	922; 672 252; 150	955; 732	955; 749; 600; 472; 318; 296; 166	456; 432; 324; 179	866; 216	933; 863	429; 220; 154; 100
		-			515	515	515	515	584; 539; 515	-	-	641; 539
OPW 11	M	+		9	-	646	670; 514	-	-	-	800; 145	-
		-			-	238	367; 293	722; 530; 238	722; 400; 293	428; 293	-	-
	H	+		6	300; 210	293	-	813; 575; 450; 378	455; 312; 232	504; 331; 268	437; 400	679; 495; 437
		-			-	-	722; 419	-	-	-	-	-
OPW 17	M	+		5	788	324	547	180	860; 644; 563; 321	309	900; 810; 419	983; 886; 846
		-			488; 449	488	362	-	221	488; 449; 221	-	488; 268; 221
	H	+		3	860; 737; 609; 514; 465; 435; 375	-	724; 582	800; 502	913; 810; 700; 574; 547; 509	567; 524	880; 800; 631	838; 604; 377
		-			-	-	321	265	321; 265	321; 265	321; 265	265
OPW 20	M	+		11	-	800; 713	710; 655; 512	900; 362	775; 524; 502; 384	818; 652; 561; 507; 306	655; 512; 287	818; 761; 649; 368; 324
		-			485; 469; 203	203	203	677; 203	203	-	-	540; 485
	H	+		9	717; 622	818; 619; 600	849; 734; 710; 605; 509	860; 717; 600; 514; 384	800; 734; 448; 386; 306; 219	818; 258	917; 791; 625;600; 386	880; 749; 710; 661; 512; 431; 362
		-			258	258	258	258	634	258	258	258

^1^ M—*Msp I*, H—*Hpa II*; ^2^ (+) appearance of a new band, and (-) disappearance of a normal band; ^3^ without AgNO_3_ and Ag-NPs, respectively.

**Table 4 plants-12-04151-t004:** Machine learning algorithm goodness-of-fit criteria for predicting callus induction (CI), embryogenic callus induction (EC), and regeneration efficiency (RE).

Traits	ML Criteria	SVM		RF		XGBoost		KNN		GP	
		Train	Test	Train	Test	Train	Test	Train	Test	Train	Test
CI ^1^	R^2^	0.281	0.098	0.402	0.244	0.551	0.515	0.076	0.068	0.539	0.443
	MSE	10.462	16.620	9.545	15.217	8.273	12.190	11.859	16.890	8.379	13.055
	MAPE	10.438	21.635	10.267	19.753	8.607	16.248	12.025	21.691	9.028	17.425
	MAD	7.761	12.118	7.876	11.287	6.677	9.523	9.298	12.172	7.006	10.332
EC	R^2^	0.383	0.574	0.436	0.719	0.648	0.393	0.144	0.432	0.595	0.706
	MSE	13.324	9.326	12.739	7.577	10.069	11.130	15.694	10.768	10.798	7.743
	MAPE	15.835	9.899	20.133	9.329	14.870	16.650	24.518	15.233	15.825	10.389
	MAD	8.665	6.923	10.472	6.165	8.334	9.983	12.547	9.169	8.522	6.916
RE	R^2^	0.526	0.505	0.502	0.422	0.671	0.461	0.145	0.236	0.659	0.525
	MSE	0.185	0.173	0.190	0.186	0.155	0.180	0.249	0.214	0.157	0.169
	MAPE	37.895	55.642	56.053	59.253	34.188	54.184	72.593	69.556	36.623	48.638
	MAD	0.131	0.154	0.161	0.167	0.121	0.159	0.201	0.191	0.126	0.138

^1^ CI: callus induction (%); EC: embryogenic callus induction (%); and RE: regeneration efficiency (number).

**Table 5 plants-12-04151-t005:** Primers and sequences used in the RAPD and CRED-RA assays.

Primer Name	Sequence (5′–3′)	Annealing Temperature (°C)
OPA 4	AATCGGGCTG	39.50
OPH 17	CACTCTCCTC	35.50
OPH 18	GAATCGGCCA	37.50
OPW 4	CAGAAGCGGA	39.50
OPW 6	AGGCCCGATG	38.50
OPW 11	CTGATGCGTG	35.50
OPW 17	GTCCTGGGTT	36.50
OPW 20	TGTGGCAGCA	44.50

## Data Availability

All data supporting the conclusions of this article are included in this article.

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
