# Peer review of "Machine Learning Analysis of the Impact of Silver Nitrate and Silver Nanoparticles on Wheat (Triticum aestivum L.): Callus Induction, Plant Regeneration, and DNA Methylation"

_plants, 2023, doi:10.3390/plants12244151_

Round 1
Reviewer 1 Report
Comments and Suggestions for Authors
Dear authors,
I've uploaded a revised version of the MS. The results are interesting but the manuscript needs to be improved. I am not convinced that the regeneration process was through somatic embryogenesis. Further evidence of somatic embryo formation must be included in the new version of the MS. Some statements are difficult to understand and part of the discussion repeats what was written in the results. The authors maus also pay attention to the way the name of the species is written - all names must be in italic.

Comments on the Quality of English LanguageI've tried to improve the quality of the English language but the authors must deeply check this point
Author Response
Responses to Comments of Reviewer 1
General Response:
First, we thank the potential reviewer for her/his valuable time and raised helpful comments and suggestions. In this step of revision, we have tried to respond to all comments and addressed all questions. We hope the revised version of manuscript gets positive feedback from you and will be acceptable for publication in the Plants journal. All revised parts have been highlighted in yellow on the manuscript.
Sincerely,
Dr. Aras Turkoglu

Reviewer 2 Report
Comments and Suggestions for Authors
I have suggested some changes in the attached PDF, which are self-explanatory.
The introduction should be reduced and to the point. Adhere to the main component of the research.
Results should be concise and focused on the salient features (results).
The analysis design (comparison of means) is not clear. Give it a second thought.
The conclusion needs a second read.

Comments on the Quality of English LanguageMinor editing of English language required.
Author Response
Responses to Comments of Reviewer 2
General Response:
First of all, we thank the potential reviewer for her/his valuable time and also raised helpful comments and suggestions. In this step of revision, we have tried to respond to all comments and addressed all questions. We hope the revised version of manuscript gets positive feedback from you and will be acceptable for publication in the Plants journal. All revised parts have been highlighted in green on the manuscript.
Comment ralated to Figure: There is no evidence that the callus are embryogenic. It seems instead an organogenic process.
Response to Comment
Dear reviewer; in vitro plant regeneration in wheat plants is generally via somatic embryogenesis. In our study, plant regeneration via indirect embryogenesis was preferred and plant growth regulators that stimulate callus and somatic embryo formation were used in this direction. For this purpose, dicamba (12 mg/L) and IAA (0.5 mg/L), which are in the auxin group, were used. Plant growth regulators to promote organogenesis (cytokinin) were not used.
When the pictures are examined carefully, embryogenic callus and somatic embryos are observed. In Figure 1A, somatic embryo formation is clearly observed in the upper left corner of the petri dish. It is also presented marked with a red colored pencil. In addition, in other pictures, somatic embryos and regenerations resulting from somatic embryos are also marked and sent.
|
Comment ralated to Figure: Letters should be inside the figures and the spaces between them must be reduced. A scale must be included;
Response to Comment
All of the figure was rearrangement based on your suggestion.

Round 2
Reviewer 1 Report
Comments and Suggestions for Authors
This is a second version of the manuscript. The authors have considerably improved the manuscript. I suggest a change in the subtitle of section 2.1 because, especially the statement "in vitro characteristics. In addition I am not convinced the calli used were embryogenic. The authors must present some evidence of the embryogenic nature of the callus.
Author Response
Responses to Comments of Reviewer 2
General Response:
First of all, we thank the potential reviewer for her/his valuable time and also raised helpful comments and suggestions. In this step of revision, we have tried to respond to all comments and addressed all questions. We hope the revised version of manuscript gets positive feedback from you and will be acceptable for publication in the Plants journal. All revised parts have been highlighted in green on the manuscript.
Comment 1; This is a second version of the manuscript. The authors have considerably improved the manuscript. I suggest a change in the subtitle of section 2.1 because, especially the statement "in vitro characteristics.
Response to Comment 1; the title was rewritten.
Comment 2;
In addition, I am not convinced the calli used were embryogenic. The authors must present some evidence of the embryogenic nature of the callus.
Response to Comment 2; dear reviwer when the pictures are examined carefully, embryogenic callus and somatic embryos are observed. In Figure 1A, somatic embryo formation is clearly observed in the upper left corner of the petri dish. It is also presented marked with a red colored arrow. In addition, in other pictures, somatic embryos and regenerations resulting from somatic embryos are also marked and sent.
